# Legislation on Disability and Employment: To What Extent Are Employment Rights Guaranteed for Persons with Disabilities?

**DOI:** 10.3390/ijerph19095654

**Published:** 2022-05-06

**Authors:** Mónica Pinilla-Roncancio, Nicolas Rodríguez Caicedo

**Affiliations:** 1School of Medicine, Universidad de los Andes, Bogotá 110111, Colombia; 2School of Governance, Universidad de los Andes, Bogotá 111711, Colombia; n.rodriguez45@uniandes.edu.co

**Keywords:** labour participation, disability, legislation, Latin America, human rights

## Abstract

Although the Convention on the Rights of Persons with Disabilities guarantees the right to employment and most countries in Latin America have signed and ratified the Convention, a large proportion of the population with disabilities still does not participate in the labour market. (1) Objective: The objective of this research was to understand how legislation in seven Latin American countries (Bolivia, Costa Rica, Chile, Colombia, Ecuador, Mexico, and Peru) has defined and enabled the inclusion of people with disabilities in the labour market. (2) Methods: We conducted a thematic analysis of the content of 34 documents and generated two thematic networks that summarise the results of the thematic analysis and represent the general relationships between the categories of analysis in each country. Using this information, we analysed the differences between countries and the advance in their legislation to fulfil the recommendations made by the Convention. (3) Results: Although all countries have enacted legislation promoting the employment rights of persons with disabilities, six of the seven countries (except Chile) have applied a medical perspective to the definition of disability in their labour legislation, thus imposing a barrier to the labour-market inclusion of this population and perpetuating the association of disability with lack of capacity to work.

## 1. Introduction

Around 70 million people in Latin America live with some form of disability [1]. As in other regions of the world, people with disabilities experience restricted access to health services and education and encounter significant barriers to participating in the labour market [2,3]. Additionally, this group experiences high levels of social exclusion and is also among the most marginalised groups, with high levels of income poverty and multidimensional poverty [4].

A person with disabilities is defined as a person who presents a functional difficulty that, together with social barriers, limits his or her participation in society [5]. Accordingly, disability is a social concept that depends directly on the context in which the person lives, and it results from socially imposed barriers that prevent a person from developing freely. Within this perspective, society plays a fundamental role at the centre of the discussion.

Various international documents mention the importance of guaranteeing the right of employment for people with disabilities. One of the most important is the Convention on the Rights of Persons with Disabilities, which contains specific articles to guarantee the rights to work, social protection, participation, health care, and education, among others. Similarly, the Convention encourages people with disabilities to access technical and vocational guidance programmes, placement services, and professional training (Article 27) [5]. In addition to the Convention, for the first time, the Sustainable Development Goals (SDGs) explicitly mention the importance of guaranteeing access to decent jobs for people with disabilities. Indeed, Goal 8.5 aims to achieve full and productive employment and decent work, and it explicitly includes people with disabilities. This is the first time that an international development target has mentioned the importance of achieving full employment for this population, with equal pay for equal work [6].

Nevertheless, in low- and middle-income countries (LMICs), it is estimated that between 80 and 90 per cent of persons with disabilities of working age are unemployed [4]. Of those who work, it is estimated that more than 60 per cent are self-employed [4], and most of them lack access to social security benefits and, in many cases, access to healthcare services or other associated benefits. Indeed, according to the International Labour Organization (ILO), fewer than 20 per cent of people with disabilities have access to social-security benefits [7]. In Latin America, the region with one of the highest levels of informal engagement in the labour market [8], a truncated welfare state, and low coverage for poor individuals working in the informal sector [9], it is estimated that more than 80 per cent of people with disabilities do not have a formal job [10]. Of those who do, 40 per cent face invisible underemployment (lower pay for the same job) or do not receive a salary.

People with disabilities in Latin America are also characterised by high levels of poverty, low levels of education, and little opportunity for social and political participation [1,3,4,11]. These characteristics increase the risk of social exclusion and exacerbate the relationship between disability and poverty [12,13]. Therefore, guaranteeing adequate access to the labour market is one of the most effective mechanisms for reducing poverty in this population [14]. However, to guarantee equal access to opportunities, it is of fundamental importance to implement reasonable adjustments. This will enable people with disabilities to have access to employment in conditions equal to those available to people without disabilities.

Seeking to contribute to the study of the inclusion of people with disabilities in the labour markets of countries in Latin America, in this article, we aim to analyse the theoretical perspectives on the labour inclusion of this group in seven countries of the region. To fulfil this objective, a thematic analysis of 34 documents was conducted. This analysis included disability and labour-market legislation, as well as public policies enacted to guarantee the employment of people with disabilities in each country.

### Employment of Persons with Disabilities: What Does the Evidence Say?

A substantial proportion of people with disabilities around the globe live in poverty. They present lower levels of education and lower levels of participation in the labour force, and they are excluded from social and political activities [4]. The relationship between poverty and disability has been described as bidirectional [13]. On the one hand, people with disabilities have a higher risk of poverty because of the barriers against their participation in society. On the other hand, low-income individuals actively face higher risks of disability, given the difficulties they face in accessing health care, as well as the different health risks to which they are exposed. One crucial factor that can reduce the risk of income poverty and multidimensional poverty of people with disabilities around the globe is access to employment and education.

Evidence in LMICs concerning access to employment for persons with disabilities is diverse. Most studies suggest that persons with disabilities face higher barriers to the formal labour market and lower labour-force participation levels [15]. Of those in work, a high proportion of the disabled population is self-employed or working in unpaid or family jobs. In addition, persons with disabilities in LMICs face a higher number of barriers to access to formal jobs that guarantee employment benefits, such as pensions, maternity leave, and sick leave, among others.

A scoping review analysing the facilitators and barriers of employment of persons with disabilities in LMICs identified that the most relevant barriers are negative attitudes in society and low levels of education and skills among persons with disabilities [16]. In this context, attitudinal barriers and discrimination become important reasons why people with disabilities do not participate in the labour market. This can affect both supply and demand aspects of employment for persons with disabilities [17]. For example, on the supply side, persons with disabilities do not receive the necessary education or the support of other household members to enable them to participate in the labour market. On the demand side, companies can discriminate against persons with disabilities and assume that the average productivity of a person with a disability is lower than that of a person without disabilities.

Comparable information across different LMICs about the employment gap between persons with and without disabilities is scarce. Mizunoya and Mitra [17], using the World Health Survey of 15 LMICs in Africa, Asia, and Latin America, estimated that in 9 of the 15 countries, persons with disabilities aged 18 to 65 presented significantly lower employment rates compared with persons without disabilities. In addition, as expected, persons with multiple disabilities presented lower employment rates than persons either without disabilities or with just one disability. In most countries, people with disabilities work in the economy’s informal sector.

Latin America includes an important percentage of people working in the informal labour market. According to the ILO [18], 53.1 per cent of the working population and 75 per cent of women in Latin America and the Caribbean work in the informal sector. Informal employment in the region is associated with low social security benefits, more precarious jobs, and income insecurity [19].

Persons with disabilities in the region also experience high levels of informal engagement in the labour market. The employment ratio of persons with and without disabilities differs between counties, with Brazil exhibiting the lowest gap (15 per cent) and Uruguay with the most significant gap (31 per cent) [20]. The disability employment gap is worse for women with disabilities in the region, who are more likely to be unemployed and, if they are working, to have a higher risk of working in the informal sector [10]. In addition, persons with disabilities in Latin America are usually unemployed or outside the labour force. Those who are employed usually receive a lower wage than a person without disabilities doing the same job. Moreover, this group faces social exclusion from the labour market, increasing their levels of income poverty and multidimensional poverty [21,22,23,24]. One study analysed the equality of employment opportunities available to persons with disabilities in six countries in Latin America (Bolivia, Chile, Colombia, Costa Rica, Mexico, and Peru). The results revealed that persons with disabilities have lower chances of employment, and important differences in the employment rate were found between groups of persons with disabilities [25].

In conclusion, there is an employment gap between persons with and without disabilities around the globe. In Latin America and the Caribbean, people with disabilities who are working are usually employed in the informal labour market. As such, they do not have access to social security benefits, a fact which increases their risk of poverty.

## 2. Materials and Methods

### 2.1. Study Design

In this analytical study, we analysed legislation related to the labour market, disability, and discrimination in seven Latin American countries. Our goal was to understand each country’s perspective on disability, the inclusion of people with disabilities in the labour market, and how, from a legal perspective, countries have implemented strategies to guarantee the right to employment and labour participation of persons with disabilities in each of the seven countries.

### 2.2. Country Selection

Countries were selected with the aim of presenting a comprehensive picture of Latin America. All selected countries have ratified the Convention on the Rights of Persons with Disabilities (Chile, 2012; Peru, 2007; Mexico, Costa Rica, and Ecuador, 2008; Colombia and Bolivia, 2009). Additionally, all selected countries, except Colombia, have ratified the optional protocol of the Convention. In addition, the selected countries are considered to be among the pioneer and intermediate countries in the development of their respective welfare states. In fact, Costa Rica and Chile inaugurated social security programmes in the 1920s; therefore, they have the longest history in terms of the implementation of their social security systems and labour-market regulations. In addition, Bolivia, Colombia, Ecuador, Mexico, and Peru began to implement their welfare states during the 1940s and have established a range of social security and protection programmes, the main objective of which is to protect vulnerable populations and to promote their participation in the labour market [26].

According to the official estimates of each country, the seven selected countries are home to around 15 million people with disabilities, which corresponds to about 15 per cent of people with disabilities in the world (see Table 1).

### 2.3. Selection of Documents

First, we identified the most relevant legislation regulating the labour market and legislation with the main objective of protecting the rights of persons with disabilities in each of the countries. Therefore, we include legal documents, public policies, and recommendations from international organisations on issues of employment and disability. After identifying each of the documents, the research team read them, and based on their content, documents were divided into five categories: “public policies on disability”, “laws on disability”, “antidiscrimination laws”, “labour market laws”, and “social protection laws” (Table A1 in Appendix A). A total of 34 documents were included in the analysis.

### 2.4. Data Analysis

#### 2.4.1. Documentary Analysis

Documentary analysis is an analytical procedure designed to review, analyse, or evaluate documents (electronic or printed) [27,28]. Like other qualitative analytical methods, documentary analysis requires that data (citations, paragraphs, testimonies, experiences, etc.) be examined and interpreted to generate meaning and knowledge and to develop empirical knowledge [29]. In addition, documentary analysis generates data from the citations in the documents, organising them into categories, topics, and examples of cases.

For this study, we regarded legal documents and national policies as documents that tell a story and provide information about the conceptualisation of different social phenomena. In this context, such documents and policies guarantee the right of inclusion in the labour market for persons with disabilities.

#### 2.4.2. Thematic Analysis

Thematic analysis is a research method that seeks to identify, organise, and analyse similarities, differences, or patterns between the documents or sources analysed. It is a method of deductive and constructive characteristics, since it is required to go beyond the data and produce an interpretation [30,31]. This analysis was conducted following four main steps:(1)Familiarisation with the data: All normative documents in each country were identified, read, and divided into different categories: documents, norms, or policies for people with disabilities and labour codes, laws on social protection, or other documents related to labour legislation in general.(2)Analytical phase: Based on the process of familiarisation, the theoretical background of the study, and the Convention on the Rights of Persons with Disabilities, the research team designed a list of categories, subcategories, and codes (Table A2, Appendix B). Using the list of categories, an inductive and deductive coding process was implemented using a preselected list of codes and allowing new categories to emerge in the data. This process was conducted by one member of the research team, and a process of checking the coding was conducted by a second member of the team.(3)Creating thematic networks: Once the coding process was finalised, the results were organised in two thematic networks. Each network represented the most relevant relationships between categories and subcategories and also reflected the theoretical understanding of important concepts behind the legislation.(4)Interpretive phase: This final phase was aimed to develop explanations based on the results. In this phase, the research team compared the results between countries and then classified countries according to their levels of agreement or disagreement on the content of the legal documents.

All analyses were conducted using AtlasTi^®^ qualitative software. 

## 3. Results

### 3.1. Thematic Networks

Two thematic networks emerged from the data. These networks originate from the concept and understanding of disability in each of the documents analysed. The first thematic network (Figure 1) captures the relationships between the human rights model to define disability and inclusion of people with disabilities in the labour market based on equal opportunities, reasonable adjustments, and non-discrimination principles.

Additionally, the first thematic network shows how labour inclusion for people with disabilities is associated with physical and sensory accessibility, reasonable adjustments, and the assurance of favourable working conditions, such as health and safety, equality in remuneration and employment, and protection from abuse or discrimination. According to the Anti-Discrimination Law of Mexico, “levelling measures are those that seek to make effective the access of all people to real equality of opportunities by eliminating physical, communicational, regulatory or other barriers that hinder the exercise of rights” (Mexico Anti-Discrimination Law, Article 15). Additionally, within this thematic network, job training programmes are included as an opportunity for companies and society to include people with disabilities, favouring their capacities and expanding their opportunities.

The second thematic network presents the relationships between social security, social assistance, the medical model of disability, and the participation of persons with disabilities in the labour market (Figure 2). This thematic network represents how most countries in the analysis have responded to the labour-market inclusion of people with disabilities. This network contradicts what is stated in Thematic Network 1. It presents a definition of disability from a medical perspective, whereby the State guarantees access to some essential goods and services but does not recognise that people with disabilities are the subjects of rights. Indeed, the State’s response takes the form of disability pensions or, in some cases, subsidies for the population with disabilities. For example, the Labour Code in Colombia mentions that “if as a consequence of a non-work-related illness or an injury or a reduction of the individual’s physical or intellectual capacity …, the worker develops a handicap that makes her incapable of earning an income higher than one-third of her income before the accident, the person will have the right to obtain a monetary benefit…” (Colombia Labour Code, Article 278).

Likewise, the second thematic network presents three primary responses designed to guarantee access to employment for people with disabilities: mandatory quotas, tax exemptions for companies, and extra incentives in public bidding processes for companies employing staff with disabilities. Some of these responses are illustrated in Peru’s Organic Law on Disability, which states: “at work, people with disabilities must represent 5% of jobs in the public sector and 3% in the private sector” (General Disability Law Peru, p. 36). In Ecuador, the Disability Law establishes tax exemptions for companies that hire people with disabilities, with a reduction of an additional 150 per cent in their income tax based on their contributions to social security for each employee with disabilities (Ecuador General Disability Law, Article 49).

### 3.2. Categories

#### 3.2.1. Right to Work

The right to work is protected for people with disabilities in specific disability legislation. For example, it is affirmed that people with disabilities have the fundamental right to access work from a perspective of human rights, equity, equal opportunity, and non-discrimination. One example is the public policy on disability in Bolivia, which states that “*work constitutes a right because it allows income generation in order to access goods and services for personal and family subsistence and to have a decent life. It generates conditions for normal social development, for the development of human potential and personal autonomy, and allows an individual to contribute to society*” (Public Policy on Disability Bolivia, p. 44). Additionally, the same legislation states that this right should be protected, guaranteeing equal remuneration for the same work. Legislation on disability states that this aspect should be enshrined under reasonable adjustment and levelling measures; this is clear in the case of Ecuador, where the disability law states that “*persons with disabilities or disabling conditions have the right to paid employment under equal conditions and not to be discriminated against*” (Ecuador Disability Law, Article 45).

#### 3.2.2. Anti-Discrimination Laws

All the analysed countries have included anti-discrimination laws in their legislation. This type of legislation contains specific sections concerning people with disabilities and their inclusion in the labour market. Anti-discrimination legislation in Mexico proscribes the following practices: “*To prohibit the free choice of employment, or to restrict the opportunities of access, retention and promotion in it; IV. Establish differences in remuneration, benefits and working conditions for equal jobs; V. Limit access and the length of training and professional training programmes*” (Mexico Anti-Discrimination Law, Article 9). The need to create conditions of equity and equality in access to employment and prohibit the discriminatory exclusion of people with disabilities is illustrated by the Peru Anti-Discrimination Law: “*The job offer and access to educational training centres may not contain requirements that constitute discrimination, cancellation or alteration of equal opportunities or treatment*” (Peru Anti-Discrimination Law, Article 1).

#### 3.2.3. Regulations on Companies

All countries except Mexico have implemented incentives to encourage companies to hire people with disabilities. Among these strategies, we found mandatory quotas to be one of the main measures. The percentage of the quotas depends on the country, ranging between two and five per cent. For example, in Colombia, “*companies must hire a percentage of people with disabilities within the existing positions, which must be published at the beginning of the fiscal year through mechanisms accessible to the population with disabilities*” (Disability Law Colombia, 2f). Chile, Ecuador, and Peru (with 3 per cent) are countries with mandatory quotas for private companies. Meanwhile, in other countries, mandatory quotas are exclusively applicable in the public sector.

Another regulation that has been implemented in the countries included in the analysis is tax exemptions granted to companies that hire people with disabilities. In Colombia, Chile, and Peru, companies that hire people with disabilities are awarded additional points in public tendering processes; for example, in Colombia, the disability law mentions that “*In any bidding process, … or direct hiring, an additional grading will be given to companies that have hired people with disabilities fulfilling all rules established by law…* ” (Colombia Disability Law).

#### 3.2.4. Education

All analysed countries have a plan or a programme to guarantee equal access to education for children and young people with disabilities. In the case of Chile, the national disability law states that “*the State will guarantee people with disabilities access to public and private establishments of the regular education system or to special education establishments, as appropriate, that receive subsidies or contributions from the State*” (Chile Disability Law, Article 34). Educational inclusion takes the form of different models and strategies, depending on the country. In Ecuador, the importance of educational inclusion is recognised, and legislation aims to guarantee this right through strategies such as specialised education, hospital classrooms, psycho-pedagogical care, and bicultural and bilingual education for deaf people, together with support and counselling units (Public Policy Disability Ecuador 2017–2021, p. 28). In Chile, the aim is to include the population with disabilities in educational institutions, promoting “*the implementation of universal design as a principle …to be installed at all educational levels and modalities*” (National Public Policy on Disability Chile 2013–2020, Article 6.3). In addition, Bolivia seeks to “*increase and apply the inclusive education approach in the educational system. In addition, it seeks to strengthen and complement inclusive education policies and the establishment of mechanisms for their monitoring and application*” (National Plan for Equality and Equalisation of Opportunities for People with Disabilities, p. 26).

### 3.3. Country Analysis

Chile, Costa Rica, and Mexico present a more significant advance regarding the labour-market inclusion of people with disabilities. These three countries have included a disability approach in their employment laws. They have recognised the importance of making reasonable adjustments and implementing actions for the inclusion of people with disabilities in the labour market. Chile has one of the most complete regulatory structures in the region. The Chilean Labour Code, Title III, refers to people with disabilities, and it encourages the use of reasonable adjustments and support services to achieve adequate work development, specifying, “*The special norms pertinent to the various classes of tasks, according to the age and sex of the workers, and to the necessary adjustments and support services that allow the worker with disabilities an adequate job performance*” (Chile Labour Code, Title III). Accordingly, the Chilean government is committed to four points for the labour-market inclusion of people with disabilities: “*(a) Promote labour practices of inclusion and non-discrimination; (b) promote the creation and design of accessible labour procedures, technologies, products and services and promote their application; (c) develop productive strategies that consider the capacities and needs of people with disabilities, which allow the generation of autonomous income; (d) disseminate the legal instruments and recommendations on the employment of people with disabilities approved by the International Labour Organization*” (Disability Law Chile, paragraph 3).

In Costa Rica, the “National Disability Policy 2011–2021”, the Labour Code, and the National Plan for Labour Inclusion were developed with the support of the United Nations Development Programme (UNDP). These documents are based on the provisions of the Convention on the Rights of Persons with Disabilities and have enabled Costa Rica to make significant progress towards the labour-market inclusion of persons with disabilities. The National Policy highlights that “*the right to work and to a decent job as a means to be able to choose independent lifestyles are vital factors for the development of equal opportunities for people with disabilities. As well as the few opportunities for insertion in the labour market and its low levels of employability …*” (National Plan for Labour Inclusion Costa Rica). Along with this, the Costa Rican government has prohibited “*all discrimination at work for reasons of age (…) [or] disability*” (Costa Rica Labour Code).

Mexico is another example of good practice. The Federal Labour Law (last reformed in 2019) and the General Law for the Inclusion of People with Disabilities (2011) have reinforced mechanisms to prevent discrimination against people with disabilities. Among these principles, we find that the Federal Law affirms that “*conditions that imply discrimination between workers on the grounds of ethnic or national origin, gender, age, disability (…) that violate human dignity may not be established*” (Federal Employment Law Mexico, Article 2). In addition, this legislation recognises the right to work, which must be based on the principle of equality. This document identifies the role of the State as to “*…promote the right to work and employment of people with disabilities in equal opportunities and equity, which gives them certainty in their personal, social and labour development*” (Federal Law of the Mexico Labour, Article 2).

In countries such as Bolivia, Colombia, Ecuador, and Peru, we identified a range of contradictions that act as barriers to the labour inclusion of people with disabilities. In these countries, we find significant progress towards guaranteeing the rights of persons with disabilities in the disability-related regulations, which closely follow the principles of the Convention. However, in laws and regulations related to the labour market, there is still legislation that reinforces the concept of disability according to a medical model, where welfare action by the State predominates.

In Bolivia, the Public Policy of Disability uses the social model. This document states that “cultural constructions become the main factor that affects and violates the human rights of people with disabilities. The lack of information and knowledge leads to stereotypes, beliefs, prejudices based on a model of society that values ‘the perfect’, ‘normality’, ‘beauty’, within highly exclusive conventional parameters” (Public Policy Disability Bolivia, 2006). However, the Bolivian General Employment Policy understands disability as an impediment to accessing employment. This law states that “*in the event of absolute and permanent disability, the victim will have the right to compensation equal to that provided in the previous article; in the event of absolute and temporary disability, compensation equal to the salary for the time that the disability lasts if it does not exceed one year, after which it will be deemed absolute and permanent, being compensated as such*” (General Labour Law Bolivia).

Colombia is one of the countries that has included the recommendations of the Convention in its legislation on disability. Indeed, both the Public Policy on Disability and the Statutory Disability Law recognise the importance of the right to work for this group. However, legislation regulating the labour market uses language that might facilitate discrimination against people with disabilities, defining this group as incapable of working. Additionally, Colombia maintains a medical model in its Labour Code, where derogatory language focuses primarily on disability. This legislation states that a person who has suffered a work-related injury or illness will have access to monetary transferences, such as: “*(a) In the event of partial permanent disability, at a sum of one (1) to ten (10) months of salary that the doctor will define once the doctor has defined the degree of disability; (b). In the event of total permanent disability, you will be entitled to a monthly disability pension equivalent to half the average monthly salary of the last year, for up to thirty (30) months and as long as the disability subsists; and (c) in the case of severe handicap, the worker will have the right to a monthly invalidity pension… for thirty months*” (Colombia Labour Code, Article 278).

In the case of Ecuador, its Labour Code mentions people with disabilities, which is not the case in most of the countries analysed. This document stipulates that “*the State shall guarantee the inclusion of people with disabilities in work, in all forms such as ordinary employment, sheltered employment or self-employment both in the public and private sectors and within the latter in national and foreign companies, as well as in other forms of production at an urban and rural level*” (Ecuador Labour Code, Article 346). However, in Ecuador, the Social Security Law continues to use the medical model of understanding disability; this law requires, “*For the purposes of this insurance [disability pension], a person is defined as disabled if as a result of an illness or physical or mental limitation,…, the person is unable to obtain a labour income at least equivalent to half of the usual remuneration that a non-disabled worker obtains in the same region*” (Ecuador Social Security Law, Article 186).

Finally, Peru presents, in Law 29973 of 2017, an important advance related to accessibility, inclusion, and labour participation of people with disabilities, whereby “*the Ministry of Labour and Employment Promotion, regional governments and municipalities incorporate the person with disabilities in their job training and updating programs, as well as in their placement and employment programs*” (Law No. 29973, 2017). In addition, it states that “*the employment services of the Ministry of Labour and Employment Promotion guarantee technical and vocational guidance to persons with disabilities, and information on job training and employment opportunities*” (Law No. 29973, 2017). However, contrary to the advance established in this law, Legislative Decree 728 presents articles that contravene the 2007 Convention. Indeed, this decree states that one of the valid causes for the termination of a labour contract is a temporary disability, which is a reason for employers to suspend a contract during the period in which the person has a disability (Legislative Decree 728, 1997).

Table 2 presents a summary of the main results of the analysis per country.

## 4. Discussion

This article aims to analyse how Bolivia, Chile, Colombia, Costa Rica, Ecuador, Mexico, and Peru have established policies to guarantee the adequate labour-market inclusion and participation of people with disabilities. In this study, we conducted a documentary and thematic analysis of 34 laws and policies on disability and the labour market. Two thematic networks were generated. The first thematic network shows the relationship between labour inclusion for the population with disabilities and accessibility, reasonable adjustments, and the guarantee of favourable conditions for work. The second thematic network highlights the connections between the medical model of disability and a welfare perspective, and it identifies the codes that fail to recognise people with disabilities as subjects of rights and active participants within the labour market, branding them instead as a passive population and the object of benefits.

The main results of this analysis revealed that only Chile has included in its Labour Code the importance of guaranteeing the right of employment for people with disabilities and the provision of reasonable adjustments. Likewise, there are actions designed to include people with disabilities based on strategies of equal opportunity, equity, increased human capacity, and the expansion of opportunities for people with disabilities. This is an important step, and it is in line with the recommendations made by the Convention on the Rights of Persons with Disabilities [5]. However, this progress in the legislation has not been reflected in the opportunities open to persons with disabilities. According to Lay-Raby et al. (2021), there are several factors affecting the participation of persons with disabilities in employment, including age, other sources of income, marital status, and education level [32].

Mexico and Costa Rica present significant advances towards guaranteeing the inclusion of people with disabilities in the labour market. These countries have implemented strategies to promote the labour inclusion of people with disabilities based on non-discrimination against this group, the provision of technical and professional training, equal opportunities, and creating favourable and accessible environments. In this context, discrimination laws become an important and relevant tool for the protection of the rights of persons with disabilities. However, people with disabilities living in countries with anti-discrimination legislation still face high levels of labour discrimination and low levels of labour participation. This is the case of Mexico and Argentina [33].

Countries such as Colombia, Ecuador, Peru, and Bolivia continue to present contradictions in the conceptualisation of disability included in labour-market and disability legislation. In these countries, the right of the disabled population to work is guaranteed in the legislation related to disability, but in the labour-market legislation, the concept of disability is presented only in terms of handicap and access to social security and disability pensions. This limits access to the formal labour market for people with disabilities, reducing their opportunities not only to be employed but also to receive social security benefits and have access to healthcare services [33].

We found that the information included in the documents can be represented by two thematic networks, which present the contradictions that were found in the legislation. For example, the first thematic network captures the association between a human rights approach, the implementation of reasonable adjustments, and the existence of aspects related to favourable work environments. Although this thematic network represents the recommendations of the Convention, most countries still understand disability as the lack of capacity to work and do not consider the advantages of hiring persons with disabilities [34]. Such advantages include those identified by a systematic review, for example, improvements in profitability and cost-effectiveness, as well as competitive advantages, including customer loyalty and satisfaction and having an inclusive work environment [35]. The second thematic network captures the understanding of disability as an incapacity to work. This network captures some of the current legal barriers that persons with disabilities face when they want to have access to the labour market. For example, barriers related to the association between disability benefits and incapacity to work have been described in other studies as potential barriers for this group’s access to the labour market [33,36,37].

The lack of coherence in the legislation regulating the labour market and individual aspects, such as low levels of education [4] and companies refusing to implement reasonable adjustment for persons with disabilities, [34] are among the factors that increase the risks of social exclusion of persons with disabilities. In addition, the lack of coherence contributes to the high percentage of people with disabilities subsisting outside the labour market. Concerning the latter aspect, evidence from Argentina suggests that 14 per cent of the population with disabilities in the country did not participate in the labour market because they believed that they would not be offered a job [38]. A similar result was found in a study conducted in Chile, where more than 40 per cent of the population with disabilities testified that they would like to go back to work if the conditions conducive to their proper participation existed [38].

Different responses to the labour-market inclusion of people with disabilities were found in each of the countries analysed. In this respect, Chile is the country that has included the most significant number of strategies. In fact, within its labour-market legislation, the importance of providing reasonable adjustments, job-training processes, and work quotas, as well as improving work environments, is mentioned. Thus, Chile is the only country where the proposals enshrined in its disability policies and laws are reflected in the disability chapter embodied in its labour code. However, this country still faces challenges to the expansion of strategies to promote labour inclusion for this group. Although the normative and legal framework of the country can serve as an example for other countries in the region, Chile still presents a low level participation by people with disabilities in the labour market when the levels of opportunities to employment are analysed. Pinilla-Roncancio and Gallardo (2022) found that persons with disabilities have lower levels of participation compared with persons without disabilities [25].

The International Labour Organization recommends different mechanisms for public and private companies to protect the employment rights of persons with disabilities, including quotas and tax incentives [7]. All the countries analysed, except for Mexico, have implemented mandatory labour quotas for public companies to guarantee the hiring of people with disabilities. However, these countries do not have mechanisms to implement, monitor, and verify compliance with this regulation as a form of labour inclusion. Furthermore, tax incentives to encourage the employment of people with disabilities and incentives incorporated in public bidding processes do not always consider prejudices when hiring people with disabilities, an aspect that can create barriers to the implementation of these processes and disincentivise the labour participation of this group.

In this article, a range of contradictions were identified that make it difficult to implement labour inclusion policies for people with disabilities. These contradictions are characterised by normative lags in which people with disabilities are not recognised as a working population, and outdated language such as “invalid”, “disabled”, or “sick” is used, along with lags on the conceptual level by which the rights of people with disabilities and the progress made in national and international disability regulations are not recognised. Additionally, countries have not aligned general policies and laws with a disability perspective and with what is recommended in the Convention on the Rights of Persons with Disabilities and the SDGs.

## 5. Limitations

Although the findings of this article reveal important normative contradictions that affect the labour-market inclusion of persons with disabilities in the seven countries analysed, the results should be considered in the light of the following limitations. First, the documents analysed in each country represent national perspectives; we did not include legislation enacted at the local level (municipalities, provinces, or regions within countries). Second, we included the most relevant legal documents; however, some countries might have other legal documents that might expand the legal framework that we analysed in this article.

## 6. Conclusions

In this study, we analysed legislation from seven countries in Latin America and aimed to understand, from a legislative perspective, the main strategies implemented to ensure the inclusion of people with disabilities in the labour market. The findings revealed that Chile is the only country that has applied a human rights perspective to define and respond to disability in labour-market legislation. All countries included in the study have signed and ratified the Convention on the Rights of Persons with Disabilities and currently have legislation to guarantee the rights of people with disabilities; however, there is still a contradiction between legislation on disability and legislation regulating the labour market. This limits the potential impact that disability legislation has on the inclusion of this group and, in some cases, increases their social exclusion. It is fundamentally important for all countries to harmonise legislation on disability with legislation governing other areas, such as employment, education, health care, and social protection.

## Figures and Tables

**Figure 1 ijerph-19-05654-f001:**
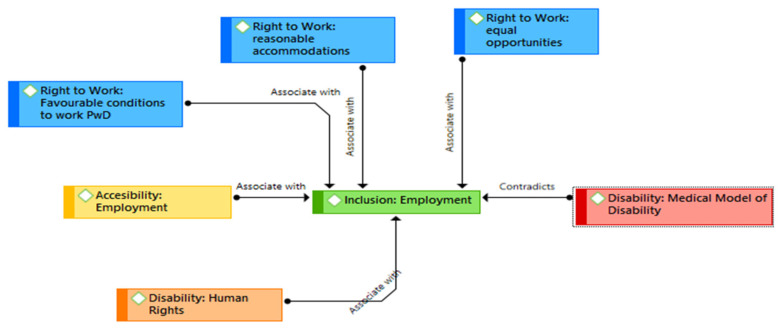
Thematic Network 1. Source: Authors’ own elaboration. PwD: people with disabilities.

**Figure 2 ijerph-19-05654-f002:**
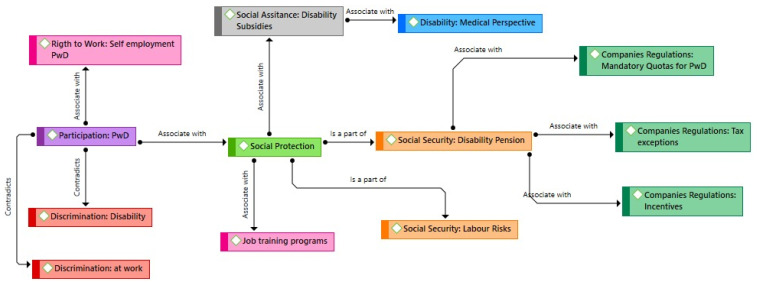
Thematic Network 2. Source: Authors’ own elaboration. PwD: people with disabilities.

**Table 1 ijerph-19-05654-t001:** Percentage of people with disabilities in selected countries.

Country	Number of People with Disabilities	Percentage	Percentage of People with Disabilities Who Are Employed
Bolivia	388,199	3.2%	11.3% ^a^
Colombia	3,500,000	7.1%	12.0% ^b^
Chile	2,836,818	16.7%	13.0% ^c^
Ecuador	485,325	2.8%	14.4% ^d^
Costa Rica	670,640	18.2%	43.0% ^e^
Mexico	7,650,000	23.4%	39.1% ^f^
Peru	3,351,919	10.3%	43.0% ^g^

^a^ National Institute of Statistics, 2012; SIPRUNPCD and the IBC, 2017. ^b^ National Administrative Department of Statistics, 2018. ^c^ II National Study on Disability, 2017. ^d^ Ministry of Health, 2020. ^e^ National Institute of Statistics and Censuses, 2018. ^f^ MCS-ENIGH, 2014. ^g^ National Statistics Office, 2017.

**Table 2 ijerph-19-05654-t002:** Summary of the main results per country.

	Bolivia	Colombia	Chile	Costa Rica	Ecuador	Mexico	Peru
Classification Welfare State	Intermediate	Intermediate	Pioneer	Pioneer	Intermediate	Intermediate	Intermediate
Perspective on disability in labour regulation	Incapacity to work	Incapacity to work	Human rights	Human rights	Incapacity to work	Human rights	Incapacity to work
Existence of a public policy on disability	Yes	Yes	Yes	Yes	Yes	No	Yes
Protection of the right to work	The Political Constitution of Bolivia (2009)	The Statutory Law on Disability (2013)	The Labour Inclusion Law in Chile (2018)	Law 7600 (1996)	The Organic Law on Disability (2012)	Political Constitution of Mexico (1917)	The Labour Inclusion Law (2016)
Anti-Discrimination Law	Yes	No	Yes	No	Yes	Yes	Yes
Regulation of Employers	Quotas	Quotas	Quotas	Quotas	Quotas	No	Quotas
	Tax exemptions	Tax exemptions	Tax exemptions		Tax exemptions		Tax exemptions
	Incentives for hiring	Incentives for hiring	Incentives for hiring		Incentives for hiring		Incentives for hiring
Skill Training Programmes	Tabaja Perú- Programa Factor Trabajo (Programme Factor Employment)	Pacto Productividad (Productivity Pact)	Fundación Avanzar-Agora (Avanzar-Agora Foundation)	Empleate (Employ yourself)	Labour Integration Service (FENEDIF SIL)-FAD Ecuador-CEE Ecuador	Fundación PARALIFE- INCLUYEME-FHADI IP-Entrale (PARALIFE Foundation)	Programa de Inclusión laboral para Personas con Discapacidad-Agora (Labour Inclusion Programme for People with Disabilities-Agora)

Source: Authors’ own elaboration.

## Data Availability

Not applicable.

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
