# Peer review of "Legislation on Disability and Employment: To What Extent Are Employment Rights Guaranteed for Persons with Disabilities?"

_ijerph, 2022, doi:10.3390/ijerph19095654_

Round 1

Reviewer 1 Report

The authors offer a well-documented analysis of laws and regulations from seven Latin American countries concerning employment for persons with disabilities.  This is an important analysis.  The authors could improve their presentation by addressing several important points.

Abstract – the background section of the Abstract is too long.  It does not include a conclusion.

Data analysis (starting line 160) – The authors define Documentary Analysis and Thematic Analysis, and outline the four steps of Thematic Analysis.  However, they do not describe how the four steps were implemented.  What did the authors do?

Results – A table to summarize the comparisons among countries would be helpful.

Discussion – (1) The authors do not integrate their discussion of results into the larger literature on employment and disability in low and middle income countries, or beyond this group of countries.  The Discussion does not include a single reference.  The authors need to indicate how the results of their research build upon and expand the existing scholarly literature.  (2) The authors do not include a statement of limitations.

Minor points

The authors’ use of specific words is confusing:

Line 90: “refer:

Line 112: “third region”

Line 117: “informality”

Line 147: “in the region”

On line 157 the authors refer to four categories, but then list five categories ("Public Policies on Disability"; "Laws of Disability"; "Antidiscrimination Laws"; "Labour Laws"; and "Social Protection Laws")

In Sections 3.2.1 and following, why are some quotations in italic and others not?

Author Response

Reviewer 1:

Abstract – the background section of the Abstract is too long.  It does not include a conclusion.

We appreciate the comments, and we have made the changes.

Data analysis (starting line 160) – The authors define Documentary Analysis and Thematic Analysis, and outline the four steps of Thematic Analysis.  However, they do not describe how the four steps were implemented.  What did the authors do?

We appreciate the comments. We have included more details about how the different steps were implemented.

Results – A table to summarize the comparisons among countries would be helpful.

We appreciate the comment and have added a comparative table to allow the reader to understand better the differences between countries.

Discussion – (1) The authors do not integrate their discussion of results into the larger literature on employment and disability in low and middle income countries, or beyond this group of countries.  The discussion does not include a single reference.  The authors need to indicate how the results of their research build upon and expand the existing scholarly literature.  (2) The authors do not include a statement of limitations.

We have changed the discussion and included a section presenting the study's limitations. The discussion starts on page 11.

The authors’ use of specific words is confusing: Line 90: “refer: Line 112: “third region” Line 117: “informality” Line 147: “in the region”

We appreciate the comments, we have made the changes, and an editor has revised the English.

On line 157 the authors refer to four categories, but then list five categories ("Public Policies on Disability"; "Laws of Disability"; "Antidiscrimination Laws"; "Labour Laws"; and "Social Protection Laws")

We appreciate the comments. We have made the changes in line 173.

In Sections 3.2.1 and following, why are some quotations in italic and others not?

We have included all quotations in italic, given that they are textual quotations.

Reviewer 2 Report

This manuscript needs to modify the following things. 

  • The title does not reflect what is analyzed at all.
  • I am not sure if the style of MDPI journal abstract is the same as that of Emerald one. Probably not.
  • There is a readability issue. Table illustration or sentence-by-sentence comparison...  There could be a variety of ways to show the results.
  • The Study Design cannot be considered a typical research design. I know describing the methodology of a descriptive study is not easy, but what this manuscript wrote is not meaningful.
  • I don't think the comparison of laws is a document analysis. The authors could directly address the method as comparison of laws. Not just documents. 
  • What we can know from the comparison of legal rhetoric cannot be  more important than what we can know from the diagnosis of the reality. 
  • The manuscript could have a clear stance in terms of disciplinary areas and methodology. Legal study, public policy, sociology ... 
  • The manuscript fails to deliver how the seven countries differ from one another in main components. The descriptive analysis does not mean description itself. It does not seem analytic at all. 

Author Response

The title does not reflect what is analyzed at all.

We appreciate the comments, and we have changed the title.

I am not sure if the style of MDPI journal abstract is the same as that of Emerald one. Probably not.

We have made the changes to the abstract

There is a readability issue. Table illustration or sentence-by-sentence comparison...  There could be a variety of ways to show the results.

We have included a table comparing the results by countries and facilitate the presentation of the results

The Study Design cannot be considered a typical research design. I know describing the methodology of a descriptive study is not easy, but what this manuscript wrote is not meaningful.

We have included more details in the methodology, starting in line 137.

I don't think the comparison of laws is a document analysis. The authors could directly address the method as comparison of laws. Not just documents. 

We have included more details in the methodology to explain why this is a documentary analysis, starting from line 176.

What we can know from the comparison of legal rhetoric cannot be more important than what we can know from the diagnosis of the reality.

The comparison of legal documents allows us to understand how the legislation in each country understands and guarantees the rights of persons with disabilities. It is different from reality because in countries such as Chile. However, the legislation guarantees the right of work for persons with disabilities, the percentage of people with disabilities working is not different from other countries. Nevertheless, having legislation protecting this right is an excellent first step to understanding the social exclusion process of persons with disabilities.

The manuscript could have a clear stance in terms of disciplinary areas and methodology. Legal study, public policy, sociology ... 

We appreciate the comment and have changed the methodology to account for this suggestion.

The manuscript fails to deliver how the seven countries differ from one another in main components. The descriptive analysis does not mean description itself. It does not seem analytic at all. 

We have included a more detailed discussion and a table that presents the main differences between countries.

Round 2

Reviewer 1 Report

The authors have responded constructively to comments.  I have no further comments.

Author Response

N&A

Reviewer 2 Report

Review comments are reflected well.

Author Response

N/A